# Association of meteorological parameters with intussusception in children aged under 2 years: results from a multisite bidirectional surveillance over 7 years in India

The INCLEN Intussusception Surveillance Network Study Group

**Correspondence to**
Dr Manoja Kumar Das;
manoj@inclentrust.org

## ABSTRACT

**Objectives** The study aimed to document the association between intussusception in Indian children and meteorological parameters and examine regional variations.

**Design** A bidirectional (retrospective and prospective) surveillance between July 2010 and September 2017.

**Setting** At 20 hospitals in India, retrospective case record review during July 2010 and March 2016 and prospective surveillance during April 2016 and September 2017 were performed.

**Participants** 2161 children aged 2–24 months with first intussusception episode were included.

**Interventions** The monthly mean meteorological parameters (temperature, sunshine, rainfall, humidity and wind speed) for the study sites were collected.

**Methods** The association between monthly intussusception cases and meteorological parameters was examined at pooled, regional and site levels using Pearson ($r$) and Spearman's rank-order ($\rho$) correlation, factorial analysis of variance, and Poisson regression or negative binomial regression analyses.

**Results** The intussusception cases were highest in summer and lowest in autumn seasons. Pearson correlation analysis showed that temperature ($r=0.056$; $p<0.05$), wind speed ($r=0.134$; $p<0.01$) and humidity ($r=0.075$; $p<0.01$) were associated with monthly intussusception cases. Spearman's rank-order correlation analysis found that temperature ($\rho=0.049$; $p<0.05$), wind speed ($\rho=0.096$; $p<0.01$) and sunshine ($\rho=0.051$; $p<0.05$) were associated with monthly intussusception cases. Poisson regression analysis resulted that monthly intussusception case was associated with rising temperature (North region, $p<0.01$ and East region, $p<0.05$), sunshine (North region, $p<0.01$), humidity (East region, $p<0.01$) and wind speed (East region, $p<0.01$). Factorial analysis of variance revealed a significant seasonal difference in intussusception cases for pooled level ($p<0.05$), 2–6 months age group ($p<0.05$) and North region ($p<0.01$). Significant differences in intussusception cases between summer and autumn seasons were observed for pooled ($p<0.01$), children aged 2–6 months ($p<0.05$) and 7–12 months ($p<0.05$).

**Conclusions** Significant correlations between intussusception cases and temperature, humidity, and

### Strengths and limitations of this study

► This study included intussusception cases in children from 20 sites in India representing different geographical and climate regions.

► A relatively large number of intussusception cases in children over 7 years were analysed at pooled, regional and site levels using multiple statistical tests for consistency and triangulation.

► Consistency in the correlation between meteorological parameters and intussusception using multiple statistical tests was reassuring.

► Data for over 5 years, collected by retrospective surveillance, may be a limitation.

wind speed were observed at pooled and regional level in India. A peak in summer months was noted, which may be used for prediction, early detection and referral for appropriate management of intussusception.

## INTRODUCTION

Intussusception is the most common acute abdomen with intestinal obstruction in children, mostly occurring in infants.[1 2] The incidence of intussusception varies widely with a mean of 74 per 100 000 (range: 9–328) among infants with the peak during 5–7 months of age.[3] The available reports from India suggest that intussusception incidence varies from 17.7 to 254 per 100 000 child-years.[4 5] These children usually present with abdominal pain or excessive crying, vomiting, bloody stools and may have abdominal mass. Ultrasound is the leading diagnostic tool. The majority of the patients can be managed by reduction method and some may need surgery. The aetiology of intussusception remains unknown in the majority of cases.[6]

Intussusception has attracted attention due to its association with the rotavirus vaccine (RVV).

India introduced RVV under the universal immunisation programme in 2016 and it expanded country-wide by 2019. A recent publication by our group reported regional variation in intussusception cases among Indian children under 2 years of age.[7] More number of cases were observed during March–June (summer months), suggesting seasonal variation. Other studies from India also reported seasonal variations.[5 8–11] There is variation in the seasonality of intussusception occurrence globally. A higher number of intussusception cases have been reported during warmer months from some countries,[12–16] while no seasonal variation was reported from other countries.[17–23] Reports from China found the intussusception caseload to be associated with meteorological parameters like temperature, sunshine and humidity.[12 24] In several reports from different countries, the association between meteorological factors and intussusception remains unclear, due to various reasons, including shorter study duration, smaller sample size and no definite statistical methods to explore the association. Although seasonal variations have been observed, no association between intussusception and meteorological parameters has been reported from India.

Multisite surveillance in India documented the epidemiology and trend of intussusception in children as part of the vaccine safety surveillance linked to the RVV introduction. Using the data collected by the Indian surveillance network, this study investigated the possible association between intussusception cases and various meteorological parameters and seasonality at the site, regional and national levels.

## METHODS

### Study area and participating hospitals

This hospital-based sentinel surveillance for intussusception in under 2-year-old children at 20 tertiary care hospitals in India included both prospective and retrospective periods. This bidirectional surveillance was comprised of retrospective (1 July 2010–31 March 2016) and prospective (1 April 2016–30 September 2017) periods of data collection. The study sites were selected from four regions (North, South, East and West) of the country, ensuring a mix of public and private tertiary care hospitals (North: three public and two private; South: two public and three private; East: five public and one private; West: two public and one private) (online supplemental figure 1). The study sites represented different climate zones in India. The method of study site selection has been detailed in the protocol published.[25]

### Case definition, case selection and data collection

Children aged 2–24 months admitted to these hospitals with intussusception were eligible. For the retrospective surveillance period, the potential cases were identified using the International Classification of Diseases 9/10 or the diagnoses in the medical record section (online supplemental table 1). These case records were reviewed to identify the intussusception cases and all identified cases were recruited. For the prospective surveillance period, all age-eligible children were screened and tracked until final diagnosis, and all intussusception cases were recruited after written informed consent. Only the first intussusception cases were included in the analysis. The cases with the first episode of intussusception were identified from the history and/or from parents. For the confirmed cases, data including demography, clinical features, management and outcome were abstracted using case record forms. External experts visited the study sites to assess the protocol adherence for case selection, record retrieval and data abstraction. Data team members visited the sites and repeated the case retrieval process for the years 2014, 2015 and 2016 to assess the retrieval completeness. The data queries were clarified referring to the source documents.

### Meteorological data collection

For the study sites, monthly mean meteorological data including temperature (°C), sunshine (total hours), rainfall (mm), humidity (%) and wind speed (km/hour) were collected from the website, World Weather Online (https://www.worldweatheronline.com/). These meteorological data collected for the period July 2010 through September 2017 were available in the public domain.

### Data management and analysis

The descriptive analysis included proportions, means (with SDs) and median (with IQR). There was no definite population catchment area for the hospitals and the estimation of intussusception incidence was not possible. The calendar months were grouped into seasons (summer, rainy, autumn, winter and spring) for the individual sites and regions (online supplemental table 2). Chi-squared goodness-of-fit test was used to examine the periodicity of intussusception and monthly differences. Kruskal-Wallis H test ($\chi^2$) was performed to document the differences between >3 groups (years, months/seasons and temperature ranges: <20°C, 21°C–25°C, 26°C–30°C and >30°C). Pearson correlation ($r$) analysis explored the association between monthly intussusception cases (dependent variable) and monthly mean meteorological parameters separately (independent variables). Spearman's rank-order correlation ($\rho$) analysis was also conducted to explore whether the monthly intussusception and monthly mean meteorological parameters covary. Factorial analysis of variance (F) was used to analyse the variations in the intussusception cases during the different seasons (dependent variable: seasonal intussusception cases and independent variables: season and year). Poisson regression or negative binomial regression (β) (for parameters with wide dispersion) analyses were performed to evaluate the relationships between the monthly intussusception cases (dependent variable) and the monthly mean meteorological parameters (independent variables). The association between intussusception and meteorological parameters was examined at pooled, regional and site levels with 95%

**Table 1** Distribution and demography of children with intussusception in India

| Pooled Years | Years | | | | | | | | |
|---|---|---|---|---|---|---|---|---|---|
| | 2010* | 2011 | 2012 | 2013 | 2014 | 2015 | 2016 | 2017† | Total |
| Number of cases, n | 68 | 209 | 244 | 260 | 278 | 365 | 429 | 308 | 2161 |
| **Gender** | | | | | | | | | |
| Male, n (%) | 45 (66.2) | 142 (67.9) | 155 (63.5) | 171 (65.8) | 194 (69.8) | 243 (66.6) | 283 (66) | 198 (64.3) | 1431 (66.2) |
| Female, n (%) | 23 (33.8) | 67 (32.1) | 89 (36.5) | 89 (34.2) | 84 (30.2) | 122 (33.4) | 146 (34) | 110 (35.7) | 730 (33.8) |
| **Age groups** | | | | | | | | | |
| 2–6 months, n (%) | 24 (35.3) | 73 (34.9) | 89 (36.5) | 86 (33.1) | 91 (32.7) | 134 (36.7) | 159 (37.1) | 122 (39.6) | 778 (36) |
| 7–12 months, n (%) | 33 (48.5) | 92 (44) | 108 (44.3) | 122 (46.9) | 113 (40.6) | 135 (37) | 155 (36.1) | 117 (38) | 875 (40.5) |
| 13–24 months, n (%) | 11 (16.2) | 44 (21.1) | 47 (19.3) | 52 (20) | 74 (26.6) | 96 (26.3) | 115 (26.8) | 69 (22.4) | 508 (23.5) |
| **Regions** | | | | | | | | | |
| North, n (%) | 9 (13.2) | 19 (9.1) | 37 (15.2) | 44 (16.9) | 48 (17.3) | 61 (16.7) | 91 (21.2) | 61 (19.8) | 370 (17.1) |
| South, n (%) | 50 (73.5) | 134 (64.1) | 130 (53.2) | 149 (57.3) | 124 (44.6) | 175 (47.9) | 183 (42.7) | 136 (44.2) | 1081 (50) |
| East, n (%) | 9 (13.2) | 45 (21.5) | 61 (25) | 54 (20.8) | 84 (30.2) | 112 (30.7) | 131 (30.5) | 97 (31.5) | 593 (27.4) |
| West, n (%) | 0 (0) | 11 (5.3) | 16 (6.6) | 13 (5) | 22 (7.9) | 17 (4.7) | 24 (5.6) | 14 (4.5) | 117 (5.4) |

*The period of data collection was 6 months (July–December).
†The period of data collection was 9 months (January–September).

CI estimation. Statistical significance was considered if p<0.05. Statistical analysis was performed using STATA V.15.0 (Stata Corp, Texas, USA).

### Patient and public involvement

The study participants or the public were not involved in the design, or conduct, or reporting, or dissemination plans of our research. The findings would be relevant for care-seeking and case management for intussusception in children. The results published shall be disseminated to the public through social media and healthcare providers.

## RESULTS
### Demography data

From July 2010 through September 2017, 2279 children aged 2–24 months with intussusception were recruited. Out of these, 118 children with a history of intussusception were excluded. Thus, 2161 children with the first intussusception were analysed. The demography and regional distribution of the intussusception cases across the years are shown in table 1. There were 1431 (66.2%) boys and 730 (33.8%) girls (male–female ratio, 1.96). Majority (n=1653, 76.5%) were infants. The number of cases increased across the years with the highest (31%) jump between 2014 and 2015. During 2010–2014, children aged 7–12 months had a higher share, which changed during 2015–2017. The South region contributed the highest (50%), followed by East (27.4%), North (17.1%) and West (5.4%) regions, which were consistent across the years.

### Meteorological data

The monthly meteorological parameters for the regions are shown in online supplemental table 3. The monthly meteorological parameters for each site are given in online supplemental table 4. Across the regions, April–June and December–February had the highest and lowest temperatures, respectively. The pattern of meteorological parameters across the regions is shown in online supplemental figure 2.

### Monthly and seasonal cycles of intussusception

The Chi-squared goodness-of-fit test revealed obvious monthly ($\chi^2$-statistic=71.36; p<0.01) and seasonal differences ($\chi^2$-statistic=382.28; p<0.01) in the intussusception caseload. Also, the Kruskal-Wallis H test revealed significant differences in the intussusception cases for months (p<0.01) and seasons (p<0.01). As shown in figure 1, more intussusception cases were observed during March in the South region, June in North and East regions, all during the summer season. Not much variation in cases was observed in the West region. The intussusception cases were lowest in September and October during these years. The peak and lowest seasons were summer (March–June; n=661, 30.6% cases) and autumn (September–October; 14.5% cases), respectively (online supplemental figure 3).

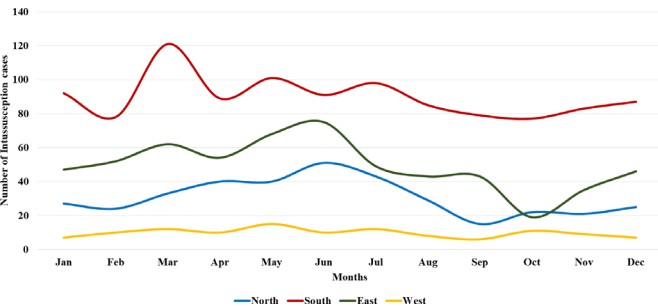

**Figure 1** Monthly distribution of intussusception cases in children during 2010–2017.

**Table 2** Pearson correlation analysis between monthly intussusception cases and monthly mean meteorological parameters

| Parameter | Pearson correlation coefficient (*r*) for meteorological parameters | | | | |
|---|---|---|---|---|---|
| | Temperature (°C) | Rainfall (mm) | Wind speed (m/s) | Humidity (%) | Sunshine (hour) |
| Pooled (n=2161) | 0.056* | −0.025 | 0.134† | 0.075† | 0.045 |
| Regions | | | | | |
| North (n=370) | 0.018 | −0.103† | −0.119† | −0.373† | 0.163† |
| South (n=1081) | 0.022 | −0.023 | 0.210† | 0.161† | −0.018 |
| East (n=593) | 0.052* | 0.072† | −0.084† | 0.109† | 0.024 |
| West (n= 117) | 0.046 | −0.019 | 0.107† | −0.029 | −0.012 |
| Age groups | | | | | |
| 2–6 months (n=778) | 0.025 | −0.022 | 0.092† | 0.040 | 0.037 |
| 7–12 months (n=875) | 0.068† | −0.012 | 0.115† | 0.069† | 0.042 |
| 13–24 months (n=508) | 0.040 | −0.030 | 0.125† | 0.076† | 0.029 |

Pearson correlation coefficient (*r*); significance two tailed.
*Significance level—p value of <0.05.
†Significance level—p value of <0.01.

## Correlation between intussusception and meteorological factors

Table 2 shows Pearson correlation analysis for pooled, regions and age groups. For pooled data, the temperature (*r*=0.056; p<0.05), wind speed (*r*=0.134; p<0.01) and humidity (*r*=0.075; p<0.01) had significant association with intussusception. At regional level, intussusception cases had association with temperature for East region (*r*=0.052; p<0.05); with rainfall for North (*r*=−0.103; p<0.01) and East regions (*r*=0.073; p<0.01); with wind speed for all regions (*r*=-0.084–0.21; p<0.01); with humidity for three regions except West region (*r*=-0.373–0.161; p<0.01); and with sunshine for North region (*r*=0.163; p<0.01). For the age groups, intussusception cases had association with temperature for 7–12 months (*r*=0.068; p<0.01), with wind speed for all age groups (*r*=0.092–0.125; p<0.01), with humidity for 7–12 months (*r*=0.069; p<0.01) and 13–24 months (*r*=0.076; p<0.01) age groups. The Pearson correlation analysis for intussusception cases with the meteorological parameters for the study sites is given in online supplemental table 5. Significant correlations for intussusception cases with temperature were observed for three sites, with rainfall for two sites, with wind speed and humidity for five sites each, and with sunshine for three sites.

Table 3 shows Spearman's rank-order correlation analysis for pooled, regions and age groups. For pooled data, the temperature (*ρ*=0.049; p<0.05), wind speed (*ρ*=0.096; p<0.01) and sunshine (*ρ*=0.051; p<0.05) were associated with intussusception. At regional level, intussusception

**Table 3** Spearman's rank-order correlation analysis between monthly intussusception cases and monthly mean meteorological parameters

| Parameter | Spearman's rank-order correlation coefficient (*ρ*) for meteorological parameters | | | | |
|---|---|---|---|---|---|
| | Temperature (°C) | Rainfall (mm) | Wind speed (m/s) | Humidity (%) | Sunshine (hour) |
| Pooled (n=2161) | 0.049* | 0.012 | 0.096† | 0.005 | 0.051* |
| Regions | | | | | |
| North (n=370) | 0.110* | −0.014 | 0.183† | −0.114* | 0.119* |
| South (n=1081) | 0.028 | −0.031 | −0.024 | −0.027 | 0.036 |
| East (n=593) | 0.061 | −0.042 | 0.109† | −0.102* | 0.038 |
| West (n= 117) | 0.052 | −0.044 | 0.000 | −0.067 | 0.076 |
| Age groups | | | | | |
| 2–6 months (n=778) | 0.035 | 0.031 | 0.063† | 0.007 | 0.044 |
| 7–12 months (n=875) | 0.064† | 0.036 | 0.103† | 0.049* | 0.048* |
| 13–24 months (n=508) | 0.028 | 0.019 | 0.106† | 0.043 | 0.016 |

Spearman's rank-order correlation coefficient (*ρ*); significance two tailed.
*Significance level—p value of <0.05.
†Significance level—p value of <0.01.

cases had association with temperature for North region ($\rho$=0.110; p<0.05); with wind speed for North ($\rho$=0.183; p<0.01) and East ($\rho$=0.109; p<0.01) regions; with humidity for North ($\rho$=−0.114; p<0.05) and East ($\rho$=−0.102; p<0.05) regions; and with sunshine for North region ($\rho$=0.119; p<0.05). For the age groups, intussusception cases had association with temperature for 7–12 months ($\rho$=0.064; p<0.01), with wind speed for all age groups ($\rho$=0.063–0.106; p<0.01), with humidity ($\rho$=0.049; p<0.05) and sunshine ($\rho$=0.048; p<0.05) for 7–12 months age group. The Spearman's rank correlation analysis for intussusception cases with the meteorological parameters for the study sites is given in online supplemental table 6. Significant correlations for intussusception cases with temperature were observed for one site, with rainfall for two sites, with wind speed for three sites, with humidity for five sites and with sunshine for two sites.

Regression analyses (Poisson regression and negative binomial regression) revealed significant association between intussusception cases and temperature for pooled level ($\beta$=0.026; 95% CI 0.011 to 0.041; p<0.01), North ($\beta$=0.027; 95% CI 0.007 to 0.047; p<0.01) and East ($\beta$=0.029; 95% CI 0.001 to 0.057; p<0.05) regions; wind speed for pooled level ($\beta$=0.082; 95% CI 0.061 to 0.103; p<0.01), North ($\beta$=0.208; 95% CI 0.101 to 0.315; p<0.01) and East ($\beta$=0.091; 95% CI 0.042 to 0.141; p<0.01) regions; humidity for pooled level ($\beta$=0.011; 95% CI 0.007 to 0.016; p<0.01) and East ($\beta$=−0.021; 95% CI −0.035 to −0.007; p<0.01) region; and sunshine for pooled level ($\beta$=0.002; 95% CI 0 to 0.003; p<0.01) and North ($\beta$=0.004; 95% CI 0.002 to 0.007; p<0.01) region (table 4). Associations between intussusception cases and temperature and humidity were evident across different age groups (p<0.01). The association was also significant at study sites level for temperature (three sites), humidity (six sites) and sunshine (five sites) (online supplemental table 7).

The monthly average temperatures at the sites and regions varied widely: 14°C–35°C in North, 23°C–31°C in South, 20°C–32°C in East and 22°C–31°C in West regions. $\chi^2$ goodness-of-fit test showed significant difference at 20°C–30°C for pooled, North and South regions (p<0.01), indicating that intussusception cases occurred more during this temperature (online supplemental table 8/13). Kruskal-Wallis H test found significant variation in intussusception cases between different temperature categories for pooled ($\chi^2$=15.933; p<0.01), North ($\chi^2$=21.054; p<0.01) and South ($\chi^2$=50.625; p<0.01) regions (online supplemental table 9/14). The number of intussusception cases increased when the mean daily temperature and the wind speed increased and mean daily humidity decreased at the pooled level and for North and East regions (online supplemental figures 4–8).

### Correlation between intussusception, seasons and age groups

Factorial analysis of variance revealed significant difference in seasonal intussusception cases at pooled level (F=1.442; p<0.05), for 2–6 months age group (F=1.579; p<0.05) and for North region (F=1.809; p<0.01) (online

supplemental tables 8 and 10 & 9 and 11). Multiple comparisons revealed significant difference in intussusception cases between summer and autumn seasons (mean difference=0.569; 95% CI 0.11 to 1.03; p<0.01) at the pooled level (online supplemental table 10 and 12). Significant differences in intussusception cases between summer and monsoon (mean difference=0.165; 95% CI 0.04 to 0.29, p<0.01), summer and autumn (mean difference=0.179; 95% CI 0.03 to 0.32, p<0.01), spring and monsoon (mean difference=0.276; 95% CI 0.09 to 0.46, p<0.01), and spring and autumn (mean difference=0.290; 95% CI 0.09 to 0.49, p<0.01) were observed for North region (online supplemental table 11 and 13). Significant differences in intussusception cases were observed between summer and autumn for children aged 2–6 months (mean difference=0.195; 95% CI 0.01 to 0.38, p<0.05) and for children aged 7–12 months (mean difference=0.231; 95% CI 0.01 to 0.45, p<0.05) (online supplemental table 12 and 14).

### DISCUSSION

The seasonal influence on intussusception cases has been proposed, but with inconsistent findings from different countries and different parts of India. This study is the first to explore the association between intussusception and monthly mean meteorological parameters (temperature, rainfall, wind speed, humidity and sunshine) in India. This study, comprising 2161 children with the first intussusception from 20 sites in India, documented monthly and seasonal variations. Intussusception case peaks were observed during summer months (March–June) and troughs during autumn months (September–October). There was a positive correlation between intussusception cases and meteorological parameters like temperature, humidity, wind speed, and sunshine at pooled and regional levels and among children aged 7–12 months. The associations were consistent across analysis tests used (Pearson correlation, Spearman's rank correlation tests, Poisson regression or negative binomial regression). The association was more pronounced for North and East regions, where the meteorological parameter fluctuation across months, were widest. Further analysis using factorial analysis of variance also demonstrated seasonal inclination, peaking in the summer season (March–June) in India and significant difference from the autumn season (with lowest cases) (p<0.01). A significant difference in the number of intussusception cases between the temperature ranges (<20°C, 21°C–25°C, 26°C–30°C and >30°C) was observed at pooled and regional (North and South regions) levels, which indicated a rise in intussusception cases with the rise in monthly mean temperature. The intussusception cases increased with the rise in daily temperature and wind speed and fall in humidity.

The findings are consistent with the report from China in 5994 children (aged 0–12 years) with intussusception, where more cases were documented in summer months with a significant positive association with mean monthly meteorological parameters (temperature, sunshine and

Table 4  Regression analysis of the relationship between monthly intussusception cases and monthly mean meteorological parameters

| Variables | Regression coefficient, β (95% CI) | SE, $S_\beta$ | χ2 value | P value |
|---|---|---|---|---|
| **Pooled*** | | | | |
| Temperature (°C) | 0.026 (0.011 to 0.041) | 0.008 | 12.098 | <0.01 |
| Rainfall (mm) | −0.001 (−0.003 to 0) | 0.001 | 2.462 | 0.117 |
| Wind speed (km/hour) | 0.082 (0.061 to 0.103) | 0.011 | 58.449 | <0.01 |
| Humidity (%) | 0.011 (0.007 to 0.016) | 0.002 | 21.903 | <0.01 |
| Sunshine (hour) | 0.002 (0 to 0.003) | 0.001 | 7.000 | <0.01 |
| **According to regions** | | | | |
| North region* | | | | |
| Temperature (°C) | 0.027 (0.007 to 0.047) | 0.0102 | 7.195 | <0.01 |
| Rainfall (mm) | 0.001 (−0.005 to 0.007) | 0.0029 | 0.169 | 0.681 |
| Wind speed (km/hour) | 0.208 (0.101 to 0.315) | 0.0545 | 14.556 | <0.01 |
| Humidity (%) | −0.010 (−0.020 to 0.001) | 0.0052 | 3.448 | 0.063 |
| Sunshine (hour) | 0.004 (0.002 to 0.007) | 0.0013 | 11.006 | <0.01 |
| South region* | | | | |
| Temperature (°C) | 0.010 (−0.037 to 0.058) | 0.0243 | 0.174 | 0.676 |
| Rainfall (mm) | −0.004 (−0.009 to 0.000) | 0.0024 | 3.251 | 0.071 |
| Wind speed (km/hour) | −0.011 (−0.045 to 0.023) | 0.0173 | 0.401 | 0.527 |
| Humidity (%) | −0.033 (−0.070 to 0.005) | 0.0190 | 2.949 | 0.086 |
| Sunshine (hour) | 0.002 (−0.001 to 0.004) | 0.0014 | 1.213 | 0.271 |
| East region* | | | | |
| Temperature (°C) | 0.029 (0.001 to 0.057) | 0.0141 | 4.245 | 0.039 |
| Rainfall (mm) | −0.001 (−0.004 to 0.001) | 0.0014 | 0.766 | 0.381 |
| Wind speed (km/hour) | 0.091 (0.042 to 0.141) | 0.0251 | 13.244 | <0.01 |
| Humidity (%) | −0.021 (−0.035 to −0.007) | 0.0070 | 9.188 | <0.01 |
| Sunshine (hour) | 0.002 (−6.100 to 0.005) | 0.0012 | 3.822 | 0.051 |
| West region† | | | | |
| Temperature (°C) | 0.052 (−0.02 to 0.124) | 1.987 | 0.036 | 0.159 |
| Rainfall (mm) | −0.002 (−0.006 to 0.002) | 0.855 | 0.002 | 0.355 |
| Wind speed (km/hour) | 0.022 (−0.042 to 0.087) | 0.463 | 0.032 | 0.496 |
| Humidity (%) | −0.002 (−0.016 to 0.011) | 0.105 | 0.006 | 0.746 |
| Sunshine (hour) | 0.003 (−0.001 to 0.007) | 2.429 | 0.002 | 0.119 |
| **According to age groups** | | | | |
| 2–6 months† | | | | |
| Temperature (°C) | 0.012 (−0.004 to 0.028) | 0.008 | 2.192 | 0.139 |
| Rainfall (mm) | −0.001 (−0.003 to 0.001) | 0.001 | 1.599 | 0.206 |
| Wind speed (km/hour) | 0.059 (0.037 to 0.081) | 0.011 | 28.241 | <0.01 |
| Humidity (%) | 0.006 (0.001 to 0.011) | 0.003 | 5.276 | 0.022 |
| Sunshine (hour) | 0.002 (0.000 to 0.003) | 0.001 | 4.615 | 0.032 |
| 7–12 months* | | | | |
| Temperature (°C) | 0.038 (0.018 to 0.057) | 0.010 | 14.288 | <0.01 |
| Rainfall (mm) | −0.001 (−0.003 to 0.002) | 0.001 | 0.447 | 0.504 |
| Wind speed (km/hour) | 0.08 (0.054 to 0.107) | 0.013 | 36.037 | <0.01 |
| Humidity (%) | 0.012 (0.006 to 0.018) | 0.003 | 14.660 | <0.01 |
| Sunshine (hour) | 0.002 (0.000 to 0.004) | 0.001 | 5.144 | 0.023 |

**Table 4** Continued

| Variables | Regression coefficient, β (95% CI) | SE, S$_β$ | χ2 value | P value |
|---|---|---|---|---|
| 13–24 months† | | | | |
| Temperature (°C) | 0.026 (0.006 to 0.047) | 0.010 | 6.460 | 0.011 |
| Rainfall (mm) | −0.002 (−0.005 to 0.000) | 0.001 | 3.564 | 0.059 |
| Wind speed (km/hour) | 0.102 (0.077 to 0.128) | 0.013 | 60.620 | <0.01 |
| Humidity (%) | 0.017 (0.010 to 0.024) | 0.004 | 22.599 | <0.01 |
| Sunshine (hour) | 0.002 (0.000 to 0.004) | 0.001 | 3.211 | 0.073 |

*Tests used for regression analysis: negative binomial regression.
†Tests used for regression analysis: Poisson regression.

precipitation).[12] Modelling using the data from 13 887 Chinese children with intussusception and applying the meteorological variation assumptions achieved a considerable agreement with the data collected.[24] These reports from China did not include humidity as a variable.

Studies on intussusception in children from India have reported higher cases during summer (March–July) and lowest during autumn (September–November) seasons. In three studies from South India, the highest and lowest intussusception cases were observed during March–June and September–October, respectively.[5 9 10] A multisite study from India reported the highest and lowest cases during March and September, respectively.[8] A study from North India observed the highest and lowest cases during June and September.[11] A report from our group on the retrospective surveillance observed a higher number of intussusception cases during the summer months (March–June) over the year.[7]

Seasonal patterns of intussusception occurrence have been observed from different countries. A higher number of intussusception cases were observed during the warmer months (May–October) than cooler months (November–April) in Taiwan.[14] Among Hong Kong children with intussusception, higher cases were observed during the summer months (May–July).[26] Seasonality in intussusception case occurrence was observed in Korean children with highest during warmer months (July–August) and lowest during cooler months (February).[27] Among Ethiopian children, higher intussusception cases were observed during June and lower during November.[13] Higher intussusception cases were observed during summer months and lower during winter months among South African children.[15] Among Israeli children, more intussusception cases were observed during the summer months and lower during winter.[16] No seasonal variation was also observed in some studies from different countries. Studies from Latin American countries and North America did not observe visible seasonal variation in cases.[17 23] Studies from Switzerland, France, New Zealand, Singapore and China also had no sizeable seasonal variations.[18–22]

The inconsistencies in the seasonal variations may be due to the sample size, age groups included and stratification in the analysis. Three studies that observed definite seasonal variation[12 14 24] and two that observed a statistically significant association between meteorological parameters and intussusception have a larger sample size (>5000 participants). The association observed in our study may be also due to the higher sample size. The reasons for the higher intussusception cases during summer are not known and the seasonality does not match with the rotavirus infection seasonality in India.[28] According to the reports from some developed countries (the USA and Australia), the intussusception incidence has declined over decades, and also the prevalence of prodrome infection symptoms (gastrointestinal and respiratory infections) in these children over the same period.[29–31] Further studies are needed to confirm the seasonality, association with meteorological parameters and variations in seasonality across the regions in India, along with the influences of the preventive healthcare services coverage, care-seeking practices, access to sanitation and hygiene, and changes in dietary practices.

A seasonal variation in another acute intestinal condition, acute appendicitis in children, has been reported. More acute appendicitis cases in children during summer months and lower during spring/autumn seasons have been reported from India, Pakistan, Iraq, China, Taiwan, Italy, Germany, Canada and the USA.[32–39] Among Chinese children, associations between acute appendicitis and temperature (p<0.01), humidity (p<0.01) and sunshine (p<0.01) were documented to support the seasonal variation.[35] Similar to intussusception, the appendicitis incidence has also declined in developed countries over decades, possibly due to changes in hygiene, sanitation, dietary practices, access to healthcare and reduction in infectious diseases.[40] Intussusception and acute appendicitis are intestinal problems and some of intussusceptions have associated appendicitis.[41–43] The climate factors are unlikely to directly cause intussusception or intussusception in children. But the linkage between seasonality and meteorological parameters and acute intestinal conditions appears to be plausible, which may be mediated by intestinal and non-intestinal infections (including activation of latent viral infections), mucosal inflammatory changes, lymphoid hyperplasia, or motility alterations and dietary intake changes. Although no direct relationship with the rotavirus infection could be established, higher gastrointestinal infection during summer and early rainy

seasons may be triggers for higher intussusception in the Indian context. The changes in intussusception and appendicitis incidence over time in parallel with industrialisation, economic improvement, improved access to vaccination, healthcare, and sanitation and hygiene practices suggest a possible association. Such information in the Indian and developing country context is not available. Thus, the seasonal variation in these conditions and association between meteorological parameters appear epidemiologically and biologically plausible, which needs further evaluation. We used Pearson correlation coefficient as the primary analysis to document the strength and direction of the relationship between meteorological parameters and intussusception. As a secondary analysis and checking for consistency in the findings, we used Spearman's rank-order correlation analysis to identify the strength of the association between ranked parameters and intussusception variables.

The Pearson correlation coefficient ($r$) and Spearman's correlation coefficient ($\rho$) values for different meteorological parameters were similar, although the $r$ values for temperature, rainfall, wind speed, and humidity were higher than the $\rho$ values, but had $r$ value lower for sunshine than the $\rho$ value at the pooled level. The differences between the $r$ and $\rho$ values were relatively small, which supports the consistent association between the variables. The consistency in findings from factorial analysis of variance and regression analysis with the correlation coefficients further confirms the association between meteorological parameters and intussusception.

The study had several strengths. It included a relatively large number of intussusception cases over a 7-year period recruited from 20 sites representing different regions. This study explored the association between intussusception cases and meteorological parameters at site, regional and pooled national levels. There was consistency in the correlation between intussusception cases and meteorological parameters across multiple statistical methods used, which was convincing. There are several limitations in the study. A major part of this study was retrospective. The sites belong to different climate zones and had variations in the meteorological parameters. The lack of a definite catchment population and referral pattern made it difficult to estimate the intussusception incidence rate.

In conclusion, this study documented higher intussusception cases during the summer months with a positive association between the number of cases with temperature and sunshine. Also, a significant positive correlation with humidity and wind speed was observed. The association was significant for the North and East regions with wider variations in the meteorological parameters across seasons. A higher degree of suspicion for infants with compatible clinical presentation for intussusception during the summer months may improve diagnosis and clinical care to prevent complications. The study provides evidence for predicting the intussusception pattern in children and sensitising the healthcare providers at community and peripheral facilities for early detection and quick referral of patients to minimise the surgical interventions and avoid fatalities. Further studies are needed to confirm the observation and improve the evidence base on the potential risk factors for the season variations and the aetiologies of intussusception.

**Acknowledgements** We acknowledge the support from the Ministry of Health and Family Welfare, Government of India for undertaking the study. We are thankful to the hospital administrators and the clinicians at the study site institutes, who supported and facilitated the undertaking of the study. We highly value the technical guidance and inputs provided by the members of the Technical Advisory Group: Mahesh K Agarwal and Pradeep Haldar, Ministry of Health and Family Welfare, Government of India; Satinder Aneja, Anju Seth and Archana Puri, Lady Hardinge Medical College, New Delhi; Ashok Patwari, Hamdard Institute of Medical Sciences and Research, New Delhi; Yogesh Kumar Sarin, Maulana Azad Medical College, New Delhi; Rakesh Aggarwal, Anshu Srivastava and Ujjal Poddar, Sanjay Gandhi Post Graduate Institute of Medical Sciences, Lucknow; Malathi Satyasekharan, Kanchi Kamakoti Chailds Trust Hospital, Chennai; Raju Sharma and Nirupam Madan, All India Institute of Medical Sciences, New Delhi; Jyoti Joshi and Deepak Polpakara, Immunization Technical Support Unit, Ministry of Health and Family Welfare, New Delhi; Umesh D Parashar; Centers of Disease Control and Prevention, Atlanta, USA; Jan Bonhoeffer, University Children's Hospital, Basel, Switzerland and Brighton Collaboration; Naveen Thacker, Child Health Foundation, Gandhigram; Rashmi Arora, Indian Council of Medical Research, Ansari Nagar, New Delhi; and Patrick L F Zuber and Christine G Maure, WHO, Geneva, Switzerland. We acknowledge the contribution of the research staff at the INCLEN Trust International: Harshpreet Kaur, Janvi Chaubey, Mrimmaya Das, Shweta Sharma and Vaibhav Jain. We highly appreciate the efforts made by the research staff at the study sites: Aarezo Bashir and Rafia, Sher-I-Kashmir Institute of Medical Sciences, Srinagar, Jammu and Kashmir; Prabha Shankar, Medanta—The Medicity Hospital, Gurgaon, Haryana; Anju Sharma, Maulana Azad Medical College, New Delhi; Anita Singh and Shubhanshu Srivastava, King George Medical University, Lucknow, Uttar Pradesh; Hemant Meena, Choithram Hospital, Indore, Madhya Pradesh; Pankaj Kumar and Shashi Kant, Indira Gandhi Institute of Medical Sciences, Patna, Bihar; Goutam Benia, IMS and SUM Medical College and Hospital, Bhubaneshwar, Odisha; Prasntajyoti Mohanty, SVP Post Graduate Institute of Paediatrics, Cuttack, Odisha; Asit Pradhan, MKCG Medical College, Berhampur, Odisha; Angshuman Chatterjee, Institute of Post Graduate Medical Education and Research and SSKM Hospital, Kolkata, West Bengal; S Yamuna, Andhra Medical College, Vishakhapatnam, Andhra Pradesh; Srinidhi Sudan, Apollo Hospitals, Hyderabad, Telengana; Rajesh Francis, Apollo Hospitals, Chennai, Tamil Nadu; T Easter Chandru, PSG Institute of Medical Sciences, Coimbatore, Tamil Nadu; Deepthy R, Julie and Anju Shivakumar, Government Medical College and SAT Hospital, Thiruvananthapuram, Kerala; Archit Vaidya, Grant Medical College and JJ Hospital, Mumbai, Maharashtra; Nimesh Chouksey, MP Shah Government Medical College, Jamnagar, Gujarat; Nidhi Singh, Fortis Escorts Hospital, Jaipur, Rajasthan; Mrinmoy Gohain, Gauhati Medical College, Guwahati, Assam; Arpita Bhattachrjee, Saugat Ghosh and Tanusmita Debnath, Agartala Government Medical College, Agartala, Tripura.

**Collaborators** The INCLEN Intussusception Surveillance Network Study Group: Manoja Kumar Das, Director Projects, The INCLEN Trust International, Delhi, India; Narendra Kumar Arora, Executive Director, The INCLEN Trust International, New Delhi, India; Bini Gupta, Assistant Research Officer, The INCLEN Trust International, New Delhi, India; Sharad Srivastava, Assistant Program Officer, The INCLEN Trust International, New Delhi, India; Apoorva Sharan, Research Officer, The INCLEN Trust International, New Delhi, India; Arindam Ray, Senior Program Officer, Bill and Melinda Gates Foundation, India Country Office, New Delhi, India; Ashish Wakhlu, Professor, Department of Paediatric Surgery, King George's Medical University, Lucknow, Uttar Pradesh, India; Bhadresh R Vyas, Professor, Department of Paediatrics, MP Shah Government Medical College, Jamnagar, Gujarat, India; Javeed Iqbal Bhat, Assistant Professor, Department of Paediatrics, Sher-I-Kashmir Institute of Medical Sciences, Srinagar, Jammu and Kashmir, India; Jayanta K Goswami, Professor, Department of Paediatric Surgery, Gauhati Medical College, Guwahati, Assam, India; John Mathai, Professor, Department of Paediatrics, PSG Institute of Medical Sciences, Coimbatore, Tamil Nadu, India; Kameswari K, Professor, Department of Paediatric Surgery, Andhra Medical College, Vishakhapatnam, Andhra Pradesh, India; Lalit Bharadia, Consultant Paediatric Gastroenterologist, Fortis Escorts Hospital, Jaipur, Rajasthan, India; Lalit Sankhe, Assistant Professor, Department of Community Medicine, Grant Medical College and JJ Hospital, Mumbai, Maharashtra, India; Ajayakumar MK, Professor, Department of Paediatric Surgery, Government Medical College and SAT Hospital,

Thiruvananthapuram, Kerala, India; Neelam Mohan, Consultant Paediatrics Gastroenterology, Medanta—The Medicity, Gurgaon, Haryana, India; Pradeep K Jena, Professor, Department of Paediatric Surgery, SCB Medical College, Cuttack, Odisha, India; Rachita Sarangi, Professor, Department of Paediatrics, IMS and SUM Medical College and Hospital, Bhubaneswar, Odisha, India; Rashmi Shad, Consultant Paediatrics, Choithram Hospital and Research Centre, Indore, Madhya Pradesh, India; Sanjib K Debbarma, Associate Professor, Department of Paediatrics, Agartala Government Medical College, Agartala, Tripura, India; Shyamala J, Consultant Paediatrics, Apollo Hospitals, Chennai, Tamil Nadu, India; Simmi K Ratan, Professor, Department of Paediatric Surgery, Maulana Azad Medical College, Delhi, India; Suman Sarkar, Assistant Professor, Department of Paediatrics, Institute of Post Graduate Medical Education and Research, Kolkata, West Bengal, India; Vijayendra Kumar, Professor, Department of Paediatric Surgery, Indira Gandhi Institute of Medical Sciences, Patna, Bihar, India; Narendra Behera, Professor, Department of Pediatrics, MKCG Medical College, Berhampur, Odisha, India; Rajamani G, Professor, Department of Paediatric Surgery, Coimbatore Medical College, Coimbatore, Tamil Nadu, India; Anand P Dubey, Professor, Department of Paediatrics, Maulana Azad Medical College, Delhi, India; Atul Gupta, Consultant Paediatric Surgery, Fortis Escorts Hospital, Jaipur, Rajasthan, India; Cenita J Sam, Professor, Department of Paediatric Surgery, PSG Institute of Medical Sciences, Coimbatore, Tamil Nadu, India; Gowhar Nazir Mufti, Assistant Professor, Department of Paediatric Surgery, Sher-I-Kashmir Institute of Medical Sciences, Srinagar, Jammu and Kashmir, India; Harsh Trivedi, Professor, Department of Paediatric Surgery, MP Shah Government Medical College, Jamnagar, Gujarat, India; Jimmy Shad, Consultant Paediatric Surgery, Apollo Hospitals, Chennai, Tamil Nadu, India; Kaushik Lahiri, Consultant, Department of Paediatric Surgery, Gauhati Medical College, Guwahati, Assam, India; Krishnaswamy R, Chief Medical Officer, Masonic Children's Hospital, Coimbatore, Tamil Nadu, India ; Meera Luthra, Consultant Paediatric Surgery, Medanta—The Medicity, Gurgaon, Haryana, India; Padmalatha P, Professor, Department of Paediatrics, Andhra Medical College, Vishakhapatnam, Andhra Pradesh, India; Rakesh Kumar, Associate Professor, Department of Paediatrics, Indira Gandhi Institute of Medical Sciences, Patna, Bihar, India; Ruchirendu Sarkar, Professor, Department of Paediatric Surgery, Institute of Post Graduate Medical Education and Research, Kolkata, West Bengal, India; Santosh Kumar A, Professor, Department of Paediatrics, Government Medical College and SAT Hospital, Thiruvananthapuram, Kerala, India; Subrat Kumar Sahoo, Associate Professor, Department of Paediatric Surgery, IMS and SUM Medical College and Hospital, Bhubaneswar, Odisha, India; Sunil K Ghosh, Associate Professor, Department of Pediatric Surgery, Agartala Government Medical College, Agartala, Tripura, India; Sushant Mane, Assistant Professor, Department of Paediatrics, Grant Medical College and JJ Hospital, Mumbai, Maharashtra, India; Arun Dash, Professor, Department of Pediatric Surgery, MKCG Medical College, Berhampur, Odisha; Bashir Ahmad Charoo, Professor, Department of Paediatrics, Sher-I-Kashmir Institute of Medical Sciences, Srinagar, Jammu and Kashmir, India; Bikasha Bihary Tripathy, Associate Professor, Department of Paediatric Surgery, IMS and SUM Medical College and Hospital, Bhubaneswar, Odisha, India; Rajendra Prasad G, Professor, Department of Paediatric Surgery, Andhra Medical College, Vishakhapatnam, Andhra Pradesh, India; Harish Kumar S, Paediatrics Radiologist, Apollo Hospitals, Chennai, Tamil Nadu, India; Jothilakshmi K, Professor, Department of Paediatrics, PSG Institute of Medical Sciences, Coimbatore, Tamil Nadu, India; Nihar Ranjan Sarkar, Associate Professor, Department of Radiology, Institute of Post Graduate Medical Education and Research, Kolkata, West Bengal, India; Pavai Arunachalam, Professor, Department of Paediatric Surgery, PSG Institute of Medical Sciences, Coimbatore, Tamil Nadu, India; Satya Sundar G. Mohapatra, Professor, Department of Radiology, IMS and SUM Medical College and Hospital, Bhubaneswar, Odisha, India; Saurabh Garge, Consultant, Paediatric Surgery, Choithram Hospital and Research Centre, Indore, Madhya Pradesh, India.

**Contributors** Study conceptualisation, study design, protocol development, training, data analysis and interpretation—MKD and NKA. Study coordination, monitoring and data cleaning—MKD, BG and AS. Data analysis, interpretation and manuscript preparation—MKD, BG and SSr. Protocol development, quality assurance and monitoring—AR. Participant recruitment and data collection—AW, BRV, JIB, JKG, JM, KK, LB, LS, MKA, NM, PKJ, RSa, RSh, SKD, JS, SKR, SSa, VK, NB, GR, APD, AG, CJS, GNM, HT, JS, KL, RK, ML, PP, RK, RSar, SKA, SKS, SKG, SM, AD, BAC, BBT, GRP, SHK, KJ, NRS, PA, GSS and SG. All authors reviewed, provided critical input and approved the final version.

**Funding** This project was supported by the Bill and Melinda Gates Foundation, USA to the INCLEN Trust International (grant number: OPP1116433).

**Disclaimer** The funder or its representative had no role in the design of the study and collection, analysis, and interpretation of data and writing the manuscript.

**Map disclaimer** The depiction of boundaries on this map does not imply the expression of any opinion whatsoever on the part of BMJ (or any member of its group) concerning the legal status of any country, territory, jurisdiction or area or of its authorities. This map is provided without any warranty of any kind, either express or implied.

**Competing interests** None declared.

**Patient consent for publication** Not required.

**Ethics approval** The study protocol was reviewed and approved by all the ethics review committees of each participating institute. The institute ethics committees of the following participating institutes approved the protocol: The INCLEN Trust International, New Delhi, India; King George's Medical University, Lucknow, Uttar Pradesh, India; MP Shah Government Medical College, Jamnagar, Gujarat, India; Sher-I-Kashmir Institute of Medical Sciences, Srinagar, Jammu & Kashmir, India; Gauhati Medical College, Guwahati, Assam, India; PSG Institute of Medical Sciences, Coimbatore, Tamil Nadu, India; King George Hospital, Andhra Medical College, Vishakhapatnam, Andhra Pradesh, India; Fortis Escorts Hospital, Jaipur, Rajasthan, India; Grant Medical College & JJ Hospital, Mumbai, Maharashtra, India; Government Medical College & SAT Hospital, Thiruvananthapuram, Kerala, India; Medanta—The Medicity, Gurgaon, Haryana, India; SCB Medical College, Cuttack, Odisha, India; IMS & SUM Medical College & Hospital, Bhubaneswar, Odisha, India; Choithram Hospital and Research Centre, Indore, Madhya Pradesh, India; Agartala Government Medical College, Agartala, Tripura, India; Apollo Hospitals, Chennai, Tamil Nadu, India; Maulana Azad Medical College, Delhi, India; Institute of Post Graduate Medical Education and Research, Kolkata, West Bengal, India; Indira Gandhi Institute of Medical Sciences, Patna, Bihar, India; Apollo Hospital, Hyderabad, Telangana, India; and MKCG Medical College, Berhampur, Odisha, India.Confidentiality in data handling was maintained. The participants under the prospective component were recruited after written informed consent from the parent/legal guardian.

**Provenance and peer review** Not commissioned; externally peer reviewed.

**Data availability statement** All data are available with the investigators and can be provided by the corresponding author upon reasonable request.

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
