## [Reviewer comments · BMJ Open]

ARTICLE DETAILS

TITLE (PROVISIONAL)	Association of meteorological parameters with intussusception in children aged under-two years: Results from a multisite bidirectional surveillance over seven years in India
AUTHORS	Study Group, The INCLEN Intussusception Surveillance; Das, Manoja

VERSION 1 – REVIEW

REVIEWER	Wan-liang Guo Children' s hospital of Soochow University, China
REVIEW RETURNED	28-Aug-2020

GENERAL COMMENTS	Dr. Manoja Kumar Das et al. explored the association 218 between intussusception in Indian children and meteorological parameters. Data for 2161 children aged 2-24 months with primary intussusception from a multisite (20 sites) bidirectional (retrospective and prospective) surveillance in India during July 2010 to September 2017 were used. The association between monthly intussusception cases and meteorological parameters were examined at pooled, regional and site levels using Pearson's and Spearman's rank-order correlation, factorial analysis-of-variance and Poisson regression or negative binomial regression analyses. They concluded that a seasonal variation in monthly intussusception was observed in India, with a peak in summer months and more evident for three regions. Monthly intussusception cases were positively associated with the monthly mean meteorological parameters, especially temperature. I have some major concerns 1. Data for 2161 children aged 2-24 months with primary intussusception from a multisite (20 sites) bidirectional (retrospective and prospective) surveillance in India during July 2010 to September 2017 were used. And only 2161 children aged 2-24 months with primary intussusception were included. Does these data can be correctly representative the occurrence of intussusception in India?2. Spearman's rank-order correlation analysis was also conducted to explore whether the monthly intussusception and monthly mean meteorological parameters covary. Factorial analysis-of variance was used to analyse the variations in the intussusception cases during the different seasons (dependent variable: seasonal intussusception cases and independent variable: season and year). Poisson regression or negative binomial regression (for parameters with wide dispersion) analyses was performed to evaluate the relationships between the monthly intussusception cases (dependent variable) and the monthly mean meteorological parameters (independent variables).
---

	Many methods were used to explore the association between mean meteorological parameters and intussusception. Which statistic methods is the best? The author should discuss it in discussion. 3. What is the clinical significance of this manuscript? Can we use it to predict the occurrence of intussusception? 4. July 2010 to September 2017 is over 7 years and there are many Spelling mistakes. Please check it in whole manuscript carefully.
--	--

REVIEWER	Dae Yong Yi Chung-Ang University Hospital, South Korea
REVIEW RETURNED	24-Sep-2020

GENERAL COMMENTS	This is very interesting as a study that confirming the occurrence of intussusception according to meteorological parameters in India. Moreover, India is a country with a large area, and because there are differences in distance between South and North, and East and West, the results of regional and climatic differences are meaningful. 1. There have already been many reports of seasonal variation in intussusception in other countries, as the authors have said. The authors have explained the correlation between climate and intussusception through statistical methods, but as the authors are aware of, it is difficult to correct various factors and the causal relationship is not clear. Therefore, the reason for such a tendency is to be supplemented in the discussion. I think that the authors only talked about the association with appendicitis. The reasons described in line 516-521 should have been explained in more detail. As the authors know, changes such as temperature, humidity, sunshine do not cause intussusception directly, but may be due to causes such as infection associated with that seasonal differences. Indeed, many studies on the incidence of intussusception over a long period of time have shown that it decreases with industrialization or improved access to medical care. It is necessary to explain the association with seasonal infection patterns, or to analyze various factors such as regional economic factors and medical accessibility. 2. In 2010, the number of patients is exceptionally small, even considering that the survey was conducted for only six months. What do you think is the reason? 3. Also, the authors said that the large number was one of the strengths of their study, and the studies that the authors referenced showed more than 5000 sample sizes. However, a recent study in South Korea involved 25,000 people. (Jo et al. Characteristics of intussusception among children in Korea: a nationwide epidemiological study. BMC Pediatrics (2019) 19:211) Given the size of India's total population, it doesn't seem to be a large number.
---

REVIEWER	Solomon C.C. Chen Ditmanson Medical Foundation Chia-Yi Christian Hospital, Taiwan
REVIEW RETURNED	03-Oct-2020

GENERAL COMMENTS	This article wants to explore whether the occurrence of Intussusception across India is related to climate. This is a
---

	national multi-center study, which shows the intentions of the authors. Although the information collected may not be complete, it should still be representative. Before this paper is accepted, there are some important issues that need to be clarified and improved. Q1. The last third paragraph of the DISCUSSION (LINE 509-521) tried to explain the reason for seasonal variation. What is the relationship between Rota virus infection and Intussusception? Is there any literature support? Before, we were worried that the Rota virus vaccine might cause Intussusception, but the vaccine used now has eliminated this doubt! Similarly, what is the relationship between Appendicitis and Intussusception? Is there any literature support? And Primary intussusception should only be caused by pure intussusception, not other diseases. Q2. Why only detect the correlation between Temperature and intussusception cases? Humidity has more sites than Temperature to have association, even the relevance of Wind speed and Sunshine is not lost to Temperature! What is the reason for choosing only temperature? I think Humidity may be more scientifically based and will become a new discovery in this research. Q3. In summary, the presentation of RESULTS is not bad, but the literature support and logic in DISCUSSION need to be strengthened. Q4. There are still many grammatical errors in English, please improve.
--	--

REVIEWER	Supika Kritsaneepaiboon Faculty of Medicine, Prince of Songkla University, THAILAND
REVIEW RETURNED	10-Nov-2020

GENERAL COMMENTS	Comments General The authors have submitted a study entitled “Intussusception in children under-two years, seasonal variations and associations 4 with meteorological parameters: Results from multisite bidirectional surveillance in 5 India over eight years. The strength is high number of study population. - Page 13 line 8: revise tears to years Introduction Good literature review and study objective. Reference #12 and 24 studies about relationship between the intussusception and meteorological parameter had number of studies about 6000 and 14000 How come you mentioned the small sample size in the previous studies in page 13 line 280-282 “The association between meteorological factors and intussusception remains unclear, due to various reasons including shorter study duration, smaller sample size, and no definite statistical methods to explore the association.
---

	How different are the climate and meteorological factors between India and China? Materials and methods Page 14 line 294 missing word “year” after two Results  - Good and intelligible information Discussion  - Could you please explain why the intussusception was high number in summer? Any meteorological factors such as rain falls, or temperature. Is the possible that high incidence of diarrhea in summer? In some countries, the intussusception was high in monsoon season and high rain falls. As we know that any viral infection (URI or GI tract infection) can induce lymphoid hyperplasia in terminal ileum. - Any relationship with history of recent viral infection, URI or recent Rota virus vaccination? - Page 24 line 510: revise bene to been
--	---

VERSION 1 – AUTHOR RESPONSE

REVIEWER COMMENTS

2. Reviewer: 1

Comments to the Author

Dr. Manoja Kumar Das et al. explored the association 218 between intussusception in Indian children and meteorological parameters. Data for 2161 children aged 2-24 months with primary intussusception from a multisite (20 sites) bidirectional (retrospective and prospective) surveillance in India during July 2010 to September 2017 were used. The association between monthly intussusception cases and meteorological parameters were examined at pooled, regional and site levels using Pearson’s and Spearman’s rank-order correlation, factorial analysis-of-variance and Poisson regression or negative binomial regression analyses. They concluded that a seasonal variation in monthly intussusception was observed in India, with a peak in summer months and more evident for three regions. Monthly intussusception cases were positively associated with the monthly mean meteorological parameters, especially temperature.

I have some major concerns

2.1. Data for 2161 children aged 2-24 months with primary intussusception from a multisite (20 sites) bidirectional (retrospective and prospective) surveillance in India during July 2010 to September 2017 were used. And only 2161 children aged 2-24 months with primary intussusception were included. Does these data can be correctly representative the occurrence of intussusception in India?

Response: The study sites have been systematically selected from different parts of India considering the. The method of selection of the sites have been published as protocol. (Reference 25. Das MK, Arora NK, Bonhoeffer J, Zuber PLF, Maure CG. Intussusception in Young Children: Protocol for Multisite Hospital Sentinel Surveillance in India. Methods Protoc. 2018;1(2):11. Published 2018 Mar 22. doi:10.3390/mps1020011)

This the data from the network should be representative of the geographic and climatic regions in India.

2.2. Spearman’s rank-order correlation analysis was also conducted to explore whether the monthly intussusception and monthly mean meteorological parameters covary. Factorial analysis-of variance was used to analyse the variations in the intussusception cases during the different seasons

(dependent variable: seasonal intussusception cases and independent variable: season and year). Poisson regression or negative binomial regression (for parameters with wide dispersion) analyses was performed to evaluate the relationships between the monthly intussusception cases (dependent variable) and the monthly mean meteorological parameters (independent variables).

Many methods were used to explore the association between mean meteorological parameters and intussusception. Which statistic methods is the best? The author should discuss it in discussion.

Response: We have responded to similar comment from the Editor above. We would like to submit again the following responses for consideration.

We used Pearson's correlation coefficient as the primary analysis to document the strength and direction of relationship between the meteorological parameters and intussusception.

As secondary analysis and checking for the consistency in the findings, we used Spearman's rank-order correlation analysis to identify strength of association between ranked parameters and intussusception variables.

The Pearson's correlation coefficient (r) and Spearman's correlation coefficient (ρ) values for different meteorological parameters were similar, although the r values for temperature, rainfall, wind speed and humidity were higher than the ρ values, but had r value was lower for sunshine than the ρ value at pooled level. The differences between the r and ρ values were relatively small, which comforts and supports the consistent association between the variables.

Factorial analysis-of-variance was used to explore the variations in the intussusception cases across seasons and years, which observed significant variation in intussusception cases across seasons at pooled level, for north region, 2-6 months age strata and seasons (summer and autumn).

Regression analysis (Poisson or negative binomial regression based on dispersion of the parameters) was done for assess the relationships between the monthly intussusception cases (dependent variable) and the monthly mean meteorological parameters (independent variables). We observed significant association between the intussusception and meteorological parameters (temperature, wind Speed, humidity and sunshine) except rainfall. These associations were consistent with the Pearson's correlation coefficient (r) and Spearman's correlation coefficient values, which further supports the consistency in association.

We have added these in the discussion section.

(Manuscript with changes marked; Page 25, lines 579-590)

2.3. What is the clinical significance of this manuscript? Can we use it to predict the occurrence of intussusception?

Response: The seasonal variations and association with the meteorological parameters suggest possible application of the information for predicting the intussusception in children and sensitizing the healthcare functionaries/providers at community and peripheral facilities for early detection and quick referral of patients to minimise the surgical interventions and also avoiding fatalities. We have added these to the discussion and conclusion. (Manuscript with changes marked; Page 23, lines 550-552 and Page 26, lines 606-609)

2.4. July 2010 to September 2017 is over 7 years and there are many Spelling mistakes. Please check it in whole manuscript carefully.

Response: We have revised the tenure to seven years. We have tried to correct the typographical and grammatical mistakes in the manuscript.

3. Reviewer: 2

Comments to the Author

This is very interesting as a study that confirming the occurrence of intussusception according to meteorological parameters in India. Moreover, India is a country with a large area, and because there are differences in distance between South and North, and East and West, the results of regional and climatic differences are meaningful.

Response: We thank the learned reviewer for encouraging feedback.

3.1. There have already been many reports of seasonal variation in intussusception in other countries, as the authors have said. The authors have explained the correlation between climate and intussusception through statistical methods, but as the authors are aware of, it is difficult to correct various factors and the causal relationship is not clear. Therefore, the reason for such a tendency is to be supplemented in the discussion. I think that the authors only talked about the association with appendicitis. The reasons described in line 516-521 should have been explained in more detail. As the authors know, changes such as temperature, humidity, sunshine do not cause intussusception directly, but may be due to causes such as infection associated with that seasonal differences. Indeed, many studies on the incidence of intussusception over a long period of time have shown that it decreases with industrialization or improved access to medical care. It is necessary to explain the association with seasonal infection patterns, or to analyze various factors such as regional economic factors and medical accessibility.

Response: Seasonal variations in intussusception cases have been reported across several countries (China, Ethiopia, Taiwan, South Africa, and Israel) and also India. Reports from some countries (Switzerland, France, New Zealand, Singapore and Latin American countries including Argentina, Brazil, Chile, Colombia, and Mexico) have reported no definite seasonal variation in the intussusception cases.

The association between the intussusception and meteorological parameters (temperature, sunshine and humidity) and seasonal variations have been documented using statistical tests only from China. We appreciate and acknowledge that it is difficult to establish any causal linkage between the meteorological parameters and intussusception. We are reporting the epidemiological association between the meteorological parameters and intussusception in Indian context and variations across the different regions. Further investigations are needed to establish the causal linkages between the potential aetiological/risk factors (like infectious, dietary, intestinal motility and any other risk factors) and intussusception.

Infectious causes have been ascribed as one of the potential cause of intussusception. Also appendicitis has been documented as a lead point trigger for intussusception. Thus, we have given appendicitis as an example to support the exploration.

The potential influence of industrialisation on intussusception has also been added to the discussion. We have added these in the discussion section.

(Manuscript with changes marked; Page 24-25, lines 559-577)

3.2. In 2010, the number of patients is exceptionally small, even considering that the survey was conducted for only six months. What do you think is the reason?

Response: The study planned collection of five years data for retrospective surveillance. The Technical Advisory Group (TAG) suggested for initiating the retrospective surveillance from July 2010. The data collection was initiated at the site institutes in January-February 2016. At the institutes usually the medical case record are kept at the hospital record section for minimum of five years and at several institutes, these records are moved to another store after five years. Thus the TAG suggested to start the surveillance from July 2010.

3.3. Also, the authors said that the large number was one of the strengths of their study, and the studies that the authors referenced showed more than 5000 sample sizes. However, a recent study in South Korea involved 25,000 people. (Jo et al. Characteristics of intussusception among children in Korea: a nationwide epidemiological study. BMC Pediatrics (2019) 19:211). Given the size of India's total population, it doesn't seem to be a large number.

Response: We don't disagree with learned reviewer about the number of intussusception cases reported in the studies from China and Korea. We would like to submit the following for consideration. The study from China (near complete data for a province) reported 5994 intussusception cases in children aged <12 years of age. It included 4808 children <2 years of age over 5 years and 5 months.

The study from Korea (near complete record of cases from the country) reported 25023 intussusception cases in children aged 0-18 years of age and the children <2 years of age were 15792 over 10 years.

Our study reports findings for 2279 children, aged <2 years, largest from India till date with regional geographical representation. Thus we have mentioned the large sample size as the strength of the study. This network collected data from 20 institutions only. Given the India's size and population this network is very small and far from complete to include all the intussusception cases. The sample included in this study is not complete but representative of the regions.

4. Reviewer: 3

Comments to the Author

This article wants to explore whether the occurrence of Intussusception across India is related to climate. This is a national multi-center study, which shows the intentions of the authors. Although the information collected may not be complete, it should still be representative. Before this paper is accepted, there are some important issues that need to be clarified and improved.

Response: We thank the learned reviewer for critical feedback.

4.1. The last third paragraph of the DISCUSSION (LINE 509-521) tried to explain the reason for seasonal variation. What is the relationship between Rota virus infection and Intussusception? Is there any literature support? Before, we were worried that the Rota virus vaccine might cause Intussusception, but the vaccine used now has eliminated this doubt!

Similarly, what is the relationship between Appendicitis and Intussusception? Is there any literature support?

And Primary intussusception should only be caused by pure intussusception, not other diseases.

Response: The available studies exploring relationship between rotavirus infection and intussusception failed to conclude any definite association as no evidence of rotavirus infection or isolation was documented in intussusception cases (including the intestinal tissues) and different seasonality pattern for the two.

(Ref: Rennels MB, Parashar UD, Holman RC, Le CT, Hwa-Gan C, Glass RI. Lack of an apparent association between intussusception and wild or vaccine virus rotavirus infection. *Pediatr Infect Dis J*. 1998; 17:924–925. DOI: 10.1097/00006454-199810000-00018

Kombo LA, Gerber MA, Pickering LK. Atreya CD and Breiman RF. Intussusception, Infection, and Immunization: Summary of a Workshop on Rotavirus *Pediatrics*. 2001; 108 (2): e37. DOI: 10.1542/peds.108.2.e37)

The current generation of rotavirus vaccines have no or low risk for intussusception and the benefit-risk ratio favours vaccine usage in routine program.

Most of the causes of intussusception are not known and are categorised as idiopathic or primary intussusception. In some intussusception cases there are comorbidities or risk factors associated like structural problems of the gastrointestinal tract, acute gastrointestinal infectious diseases, acute respiratory infectious diseases, other infectious diseases, and systemic non-infectious diseases. The prospective surveillance component of our study found any lead point in 14.6%. Although lymph node/payer's patch was the commonest, appendicitis was also documented as the lead point. (Ref: The INCLEN Intussusception Surveillance Network Study Group., Das, M.K., Arora, N.K. et al. Prospective surveillance for intussusception in Indian children aged under two years at nineteen tertiary care hospitals. *BMC Pediatr* 20, 413 (2020). <https://doi.org/10.1186/s12887-020-02293-5>)

The report from Korea reported an associated co-morbidity or trigger in 57% intussusception cases in children. Acute gastrointestinal infectious diseases were documented in 18% of these intussusceptions. Structural problems of the gastrointestinal tract were observed in 4.8% of the intussusception cases and acute appendicitis contributed to about one third of them. Thus, acute appendicitis is associated with intussusception as a trigger. (Jo, S., Lim, I., Chae, S. et al.

Characteristics of intussusception among children in Korea: a nationwide epidemiological study. *BMC Pediatr* 19, 211 (2019). <https://doi.org/10.1186/s12887-019-1592-6>)

Also we considered acute appendicitis as an example of intestinal disorder which has association with the climate factors.

4.2. Why only detect the correlation between Temperature and intussusception cases? Humidity has more sites than Temperature to have association, even the relevance of Wind speed and Sunshine is not lost to Temperature! What is the reason for choosing only temperature? I think Humidity may be more scientifically based and will become a new discovery in this research.

Response: We have explored the association for various climate/meteorological parameters including temperature, humidity, wind speed, rainfall and sunshine. The findings suggest significant correlation between temperature, humidity and wind speed as observed by you. We apologize for mentioning only temperature in the conclusions. We have revised the results, discussion and conclusions including the humidity and wind speed along with temperature.

(Manuscript with changes marked; Page 20, lines 452-461; Page 22, lines 506-507; Page 26, lines 600-611)

4.3. In summary, the presentation of RESULTS is not bad, but the literature support and logic in DISCUSSION need to be strengthened.

Response: We have revised the discussion section.

(Manuscript with changes marked; Page 21-25)

4.4. There are still many grammatical errors in English, please improve.

Response: We have tried to address the typographical and grammatical issues.

5. Reviewer: 4

Comments to the Author

Comments-General

The authors have submitted a study entitled "Intussusception in children under-two years, seasonal variations and associations 4 with meteorological parameters: Results from multisite bidirectional surveillance in 5 India over eight years. The strength is high number of study population.

Response: We thank the learned reviewer for critical feedback.

5.1. Page 13 line 8: revise tears to years

Response: Apologies, we have corrected the typographical mistake.

5.2. Introduction

5.2.1. Good literature review and study objective.

Response: Thank you for the encouraging words.

5.2.2. Reference #12 and 24 studies about relationship between the intussusception and meteorological parameter had number of studies about 6000 and 14000

How come you mentioned the small sample size in the previous studies in page 13 line 280-282 "The association between meteorological factors and intussusception remains unclear, due to various reasons including shorter study duration, smaller sample size, and no definite statistical methods to explore the association.

Response: The references 12 and 24 are studies from China and included 5994 and 13887 intussusception cases, respectively. In the statement referred by reviewer, we intended to highlight the challenges with the reports from several countries that had methodological and sample size related problems to identify the potentially association between intussusception and seasonality and meteorological parameters. We have revised the statement. ((Manuscript with changes marked; Page 13, line 297)

5.2.3. How different are the climate and meteorological factors between India and China?

Response: The meteorological parameters for the Suzhou, China, where the two studies were conducted range as follows

- Monthly mean temperature: 6-8°C in winter to 32-35°C in summer
- Monthly mean rainfall: 50 mm to 600 mm
- Monthly mean wind speed: 14 kmph to 20 kmph
- Monthly mean humidity: 58% to 81%
- Monthly mean sunshine: 122- 280 hours

(Source: <https://www.worldweatheronline.com/suzhou-weather-averages/jiangsu/cn.aspx>)

These meteorological parameters are somewhat comparable to some of the north-Indian cities included in our study.

5.3. Materials and methods

5.3.1. Page 14 line 294 missing word “year” after two

Response: We have included the ‘year’. (Manuscript with changes marked; Page 140, line 311)

5.4. Results

5.4.1. Good and intelligible information

Response: Thank you.

5.5. Discussion

5.5.1. Could you please explain why the intussusception was high number in summer? Any meteorological factors such as rain falls, or temperature. Is the possible that high incidence of diarrhea in summer? In some countries, the intussusception was high in monsoon season and high rain falls. As we know that any viral infection (URI or GI tract infection) can induce lymphoid hyperplasia in terminal ileum.

Response: We have tried to address a comment from Reviewer 2 above. There is wide variation in the seasonality pattern of intussusception according to the reports from different parts of the globe. Seasonal variations in intussusception cases have been reported across several countries (China, Ethiopia, Taiwan, South Africa, and Israel) and also India. Reports from some countries (Switzerland, France, New Zealand, Singapore and Latin American countries including Argentina, Brazil, Chile, Colombia, and Mexico) have reported no definite seasonal variation in the intussusception cases. Apart from the seasonality, the only report from China reported correlation between intussusception and meteorological parameters using statistical tests and modelling approach (Reference 12 and 24). These articles have reported the association between the intussusception and meteorological parameters (temperature, sunshine and humidity) and seasonal variations.

The available reports from United States and Australia reported decline in the intussusception incidence with economic development and industrialisation, which may be due to the improvement in access to healthcare, vaccines/preventive measures, hygiene and sanitation and also reduced infections.

Although no direct relationship with the rotavirus infection could be established, higher gastrointestinal infection during summer and early rainy seasons may be triggers for higher intussusception in Indian context. Further investigations are needed to establish the causal linkages between the potential aetiological/risk factors (like infectious, dietary, intestinal motility and any other risk factors) and intussusception.

We have added these in the discussion section.

(Manuscript with changes marked; Page 23, lines 545-548; Page 24, lines 559-575)

5.5.2. Any relationship with history of recent viral infection, URI or recent Rota virus vaccination?

Response: The prospective surveillance documented prevalence of acute gastrointestinal and respiratory infection in 12.9% and 23.7% children. The information have been published. (Ref: The INCLIN Intussusception Surveillance Network Study Group., Das, M.K., Arora, N.K. et al. Prospective surveillance for intussusception in Indian children aged under two years at nineteen tertiary care hospitals. BMC Pediatr 20, 413 (2020). <https://doi.org/10.1186/s12887-020-02293-5>)

5.5.3. Page 24 line 510: revise bene to been

Response: The typo error has been corrected.

VERSION 2 – REVIEW

REVIEWER	Guo, Wan-liang the children's hospital affiliated to soochow university, radiology
REVIEW RETURNED	17-Dec-2020

GENERAL COMMENTS	Manoja Kumar Das et al. report on a series of 2161 children aged 2-24 months with primary intussusception At 20 hospitals in India, retrospective case record review during July 2010 and March 2016 and prospective surveillance during April 2016 and September 2017. Their aims to document the association between intussusception in Indian 217 children and meteorological parameters and examine the regional variations. The association between monthly intussusception cases and meteorological parameters were examined at pooled, regional and site levels using Pearson's (r) and Spearman's rank-order (ρ) correlation, factorial analysis-of-variance and Poisson regression or negative binomial regression analyses. They concluded that this study documented higher intussusception cases during the summer months with a positive association between number of cases with temperature and sunshine. Also significant positive correlation with the humidity and wind speed was observed. The association was significant for the North and East regions with wider variations in the meteorological parameters across seasons. Good study, however following needs to be taken care of:  1. 2279 children aged 2-24 months with intussusception were recruited. Out of these, 118 children had past history of intussusception, and were excluded. Why did the author exclude 118 cases in the present study? 2. How did the author define primary intussusception in the present study? 3. Page 24 . the author claimed that some of the intussusception have associate with appendicitis. To the best of my knowledge, very few intussusception in children associated with appendicitis. Is this true? 4. Page 25, "The study had several strengths; large number of intussusception cases over seven years period". large number should be "relative large sample". 5. the design of the present paper is A bidirectional (retrospective and prospective). Are there differences association intussusception in Indian children and meteorological parameters?
--

REVIEWER	Yi, Dae Yong Chung Ang University Hospital, Pediatrics
REVIEW RETURNED	12-Dec-2020

GENERAL COMMENTS	The authors have fully revised and supplemented the previously mentioned content. I think the association between Intussusception and seasonality will be different from country to country because there are various factors. Even assuming that the cause of intussusception is only infection, various factors such as the clarity of seasonal changes, economic status of the country, and hygiene will affect. Nevertheless, since intussusception is caused by structural problems that become leading points or many other causes, seasonality is thought to be inevitably different from country to country. I think you've added these parts as well. It seems that the reference I mentioned in the previous review was mentioned in the answer to another reviewer. I think you may add the literature to the reference.
---

REVIEWER	Chen, Solomon Ditmanson Medical Foundation Chia-Yi Christian Hospital, Department of Pediatrics
REVIEW RETURNED	15-Dec-2020

GENERAL COMMENTS	All my previous concerns have been well addressed. Thanks.
--

VERSION 2 – AUTHOR RESPONSE

Reviewer Reports:

Reviewer: 1

Comments to the Author:

Manoja Kumar Das et al. report on a series of 2161 children aged 2-24 months with primary intussusception At 20 hospitals in India, retrospective case record review during July 2010 and March 2016 and prospective surveillance during April 2016 and September 2017. Their aims to document the association between intussusception in Indian 217 children and meteorological parameters and examine the regional variations. The association between monthly intussusception cases and meteorological parameters were examined at pooled, regional and site levels using Pearson's (r) and Spearman's rank-order (ρ) correlation, factorial analysis-of-variance and Poisson regression or negative binomial regression analyses. They concluded that this study documented higher intussusception cases during the summer months with a positive association between number of cases with temperature and sunshine. Also significant positive correlation with the humidity and wind speed was observed. The association was significant for the North and East regions with wider variations in the meteorological parameters across seasons.

Good study, however following needs to be taken care of:

1. 2279 children aged 2-24 months with intussusception were recruited. Out of these, 118 children had past history of intussusception, and were excluded. Why did the author exclude 118 cases in the present study?

Response: In view of potential differences in the degree of association for the first and recurrent intussusception episodes with the meteorological parameters, we only considered the first intussusception episodes. Thus the 118 children with past history of intussusception were excluded from analysis.

2. How did the author define primary intussusception in the present study?

Response: We inadvertently mentioned the first intussusception episode as primary intussusception. We have corrected the primary intussusception to first intussusception episode.

(Manuscript with track changes: Page 15- line 313; Page 17- line 366; Page 21- line 468; Page 42, Legend of figure 1)

3. Page 24. The author claimed that some of the intussusception have associate with appendicitis. To the best of my knowledge, very few intussusception in children associated with appendicitis. Is this true?

Response: Yes. There are case reports of intussuscepting in children associated with appendicitis. It is uncommon, as mentioned by reviewer. We have added the following references to the statement (Ref 41-43, Page 35-36, Manuscript with track changes).

41. Atkinson G, Gay B, Naffis D. Intussusception of the appendix in children. *Am J Roentgenol.* 1976 Jun;126(6):1164–8.

42. Ozkisacik SK, Erdem AO, Coskun O, Yazici M. Small bowel intussusception together with appendicitis in childhood: A case report. *J Pediatr Surg Case Rep.* 2015 Jan;3(1):25–6.

43. Marjon L, Hull N, Thomas K. Concurrent acute appendicitis and ileocolic intussusception in a 1-year-old child. *Radiol Case Rep.* 2018 Jun;13(3):655–7.

4. Page 25, “The study had several strengths; large number of intussusception cases over seven years period”. large number should be “relative large sample”.

Response: We have modified and added the ‘relatively large sample’ in the strengths section. (Manuscript with track changes: Page 25, line 569)

5. The design of the present paper is A bidirectional (retrospective and prospective). Are there differences association intussusception in Indian children and meteorological parameters?

Response: We have analysed the data for all the years combined and not separately for the retrospective and prospective surveillance periods.

Reviewer: 2

Comments to the Author:

The authors have fully revised and supplemented the previously mentioned content.

I think the association between Intussusception and seasonality will be different from country to country because there are various factors. Even assuming that the cause of intussusception is only infection, various factors such as the clarity of seasonal changes, economic status of the country, and hygiene will affect. Nevertheless, since intussusception is caused by structural problems that become leading points or many other causes, seasonality is thought to be inevitably different from country to country. I think you've added these parts as well.

It seems that the reference I mentioned in the previous review was mentioned in the answer to another reviewer. I think you may add the literature to the reference.

Response: The Chinese studies are already part of the references (Ref 12 and 24). We have added the Korean study to the reference as suggested (Ref 27, Page 34, Manuscript with track changes).

27. Jo S, Lim IS, Chae SA, Yun SW, Lee NM, Kim SY, et al. Characteristics of intussusception among children in Korea: a nationwide epidemiological study. *BMC Pediatr.* 2019 Dec;19(1):211.

Reviewer: 3

Comments to the Author:

All my previous concerns have been well addressed. Thanks.

Response: Thank you very much.

Response to editorial comments dated Jan 28, 2021

1. When searching for an institution, Ringgold connected institutions will automatically appear in the institution list. If you're not able to find your institution you are permitted to enter your own, but that institution will not be connected to Ringgold. We suggest trying a few alternate spellings:

- Try the full name of the institution
- Is there a sub-name within the overall name? (for instance, some universities will have named departments like University of Virginia's Darden School of Business)
- Still not able to find your institution? Contact Ringgold directly to double check the institution is included in the Ringgold Database at support.ringgold.com/.

Response: The corresponding author's institution (The INCLEN Trust International, which was also the coordinating institute) is not included in the Ringgold Database. Thus we have entered the name.

2. Research Ethics Approval: Human Participants no ID number of approval

Question is: Does this study involve human participants? Yes/No

You have indicated 'Yes' to this question, Please indicate the name of the Ethics Committee(s) or Institutional Review Board(s) that approved the study, along with the number/ID of the approval(s).

Response: The study involved human participants. (Yes)

The protocol was reviewed and approved by the Institute Ethics Committee of all the participating institutes (20 participating site institutes and the coordinating institute). The names of the IECs with protocol reference number(s) are listed as Supplementary Document 2.

3. I have noticed that the name 'Das, Manoja Kumar' included in your author's list. However, upon checking the contributorship statement, I cannot find an initial that corresponds to its name. Kindly confirm.

Response: The contributorship statement revised with addition of contributions by author Manoja Kumar Das (MKD).

4. Kindly remove all your Supplementary Table in your Main Document and upload it separately under file designation "Supplementary File" in PDF Format.

Response: The list of supplementary documents have been removed from the main document. The supplementary documents (tables and figures) related to the main text have been uploaded as Supplementary Document 1.

5. Contributor ship - Please note that the statement in the ScholarOne system and main document should be the same.

Response: We have corrected the contributorship statements in the main document and ScholarOne system

VERSION 3 – REVIEW

REVIEWER	Guo, Wan-liang the children's hospital affiliated to soochow university, radiology
REVIEW RETURNED	20-Feb-2021

GENERAL COMMENTS	All my previous concerns have been well addressed. The manuscript has been fully revised and can be accepted. Thanks.
---